# Reliable uncertainty estimates in deep learning with efficient Metropolis-Hastings algorithms

**Matthias Schmal** [1] ✉ **& Patrick Mäder** [1,2,3]

Approaching problems with data-driven models often requires reliable uncertainty estimates. Bayesian neural networks can offer these for deep learning models. Without the knowledge to set informative prior distributions, sampling methods such as Hamiltonian Monte Carlo are a robust choice. However, these come with prohibitive computational costs. We study two ways to incorporate computationally light-weight Metropolis-Hastings acceptance steps into deep neural networks and stochastic gradient Hamiltonian Monte Carlo. The first method proposes noisy acceptance steps computed on batched training samples rather than the entire set during the simulation of the stochastic dynamics accepting a small minima preserving bias. The second method sacrifices bias-free sampling of Hamiltonian Monte Carlo in favor of stochastic gradient driven trajectories. While the first is analytically plausible, the second is inspired by the Hamiltonian ensemble concept. Prediction accuracy is improved by up to 5.8% over deterministic and by up to 4.3% over Bayesian approaches while still guaranteeing calibration of the predictions. We observe that sampling methods facilitate model predictions with merely a third of the ensemble while maintaining prediction accuracy. In conclusion, the methods combine efficiency and regularization of stochastic gradients, showing strong performance despite the sampling bias.

Deep Learning has become a working horse of modern machine learning. In the presence of large datasets, state-of-the-art results are achieved in image classification, segmentation, reinforcement learning, language processing and many more[1]. Throughout these applications the models match or outperform humans on the same tasks. However, deep neural networks lack uncertainty awareness[2,3]. For applications in medicine[4], autonomous driving[5] and other dependable applications[6] this is a serious drawback. A solution has the potential to unfold the field of high performing deep learning for tasks, that require information on reliability. In contrast to deep learning[1], Bayesian learning is well known for robust modeling in presence of noise and for uncertainty estimation based on parameter distributions[7,8]. It is used in

control or traditional statistical modeling and builds upon a well grounded theory[8]. The concept of Bayesian neural networks combines both methods to extract features with deep architectures and still acquire uncertainty estimations on the predictions. Even with simple uninformed prior distributions this enables complicated parameter posteriors[9] in contrast to simple post-processing[3] or Bayesian classifiers on top of a deep neural network[6,10–12]. The key to Bayesian neural networks remains the inference. Since the posterior distribution is analytically intractable, an approximation is required to estimate the posterior probability density and increased attention is required for big data applications due to practical computational limits[2,6,13]. The approximation is typically realized by either variational inference or

[1]Data-intensive Systems and Visualization Group, Technische Universität Ilmenau, Ilmenau, Thüringen, Germany. [2]Faculty of Biological Sciences, Friedrich Schiller University, Jena, Thüringen, Germany. [3]German Centre for Integrative Biodiversity Research—iDiv, Leipzig, Saxony, Germany. ✉e-mail: matthias.schmal@tu-ilmenau.de

Markov Chain Monte Carlo sampling. Variational inference uses a fixed parametric posterior that is optimized to minimize the Kullback-Leibler divergence to the true parameter distribution[2,14,15]. This approach was first published as a Bayesian neural network and uses variational formulations of backpropagation. The main limitation remains the fixed parameter posterior, which limits the diversity in the Bayesian evaluation process. Markov Chain Monte Carlo methods[16] sample from the parameter posterior, such that the collected samples allow to solve the predictive distribution with Monte Carlo integration using

$$p(y^*|x^*, S) = \int p(y^*|x^*, \boldsymbol{\theta})\, p(\boldsymbol{\theta}|S)\, d\boldsymbol{\theta}. \tag{1}$$

Here, the predictive distribution on a new data point $(y^*, x^*)$ is computed by marginalizing over the model parameters $\boldsymbol{\theta}$ that are inferred based on the dataset $S = (y_j, x_j)_J$. While deep ensembles[17] can be considered a primitive collection of optima approximation to $p(\boldsymbol{\theta}|S)$, Bayesian sampling methods converge towards the true posterior. While early Markov Chain Monte Carlo methods are very inefficient on high-dimensional problems, Hamiltonian Monte Carlo overcame these problems by utilization of a momentum variable. Its Stan package implementation is the current state-of-the-art[18–20]. The problem for deep neural networks is the required gradient of the potential energy term

$$U(\boldsymbol{\theta}) = -\sum_J \log p(y_j|x_j, \boldsymbol{\theta}) - \log p(\boldsymbol{\theta}). \tag{2}$$

Every step requires gradients on the whole dataset $S$[21]. The natural solution is to turn to stochastic gradient driven dynamical systems that allow the inclusion of step-wise additional noise terms. These simulate first or second order Langevin-dynamics, which have the posterior as their stationary distribution. Typically, the dynamics

$$\begin{cases} d\boldsymbol{\theta} = M^{-1}\mathbf{m}\, dt \\ d\mathbf{m} = -\gamma\mathbf{m}\, dt - \nabla_{\boldsymbol{\theta}}\tilde{U}(\boldsymbol{\theta})\, dt + M^{1/2}\sqrt{2\gamma T}\, dW \end{cases} \tag{3}$$

with $M \in \mathbb{R}^{d \times d}$ invertible and $W$ a standard Wiener process with variance $I$ are computed with

$$\nabla_{\boldsymbol{\theta}}\tilde{U}(\boldsymbol{\theta}) = -\frac{|S|}{|B|}\sum_{j\in B}\nabla_{\boldsymbol{\theta}}\log p(y_j|x_j, \boldsymbol{\theta}) - \nabla_{\boldsymbol{\theta}}\log p(\boldsymbol{\theta}). \tag{4}$$

This state of the dynamical system (3) consists of the model parameters and a momentum state $m$. We orientate our notation on[22–24]

### Table 1 | Overview on discretizations of Eq. (3)

**Euler-Maruyama**
$\mathbf{m}_{t+1} = (1 - \epsilon\gamma)\mathbf{m}_t - \epsilon\nabla_{\boldsymbol{\theta}}\tilde{U}(\boldsymbol{\theta}_t) + \sqrt{2\epsilon\gamma TM}\,\mathbf{w}_t$
$\boldsymbol{\theta}_{t+1} = \boldsymbol{\theta}_t + \epsilon M^{-1}\mathbf{m}_{t+1}$

**Skew-symmetric Integrator**
$\boldsymbol{\theta}_{t+\frac{1}{2}} = \boldsymbol{\theta}_t + \frac{\epsilon}{2}M^{-1}\mathbf{m}_t$
$\mathbf{m}_{t+1} = \frac{(1-\frac{1}{2}\epsilon\gamma)\mathbf{m}_t - \epsilon\nabla_{\boldsymbol{\theta}}\tilde{U}\left(\boldsymbol{\theta}_{t+\frac{1}{2}}\right) + \sqrt{2\epsilon\gamma TM}\,\mathbf{w}_t}{(1+\frac{1}{2}\epsilon\gamma)}$
$\boldsymbol{\theta}_{t+1} = \boldsymbol{\theta}_{t+\frac{1}{2}} + \frac{\epsilon}{2}M^{-1}\mathbf{m}_{t+1}$

**Symmetric OBABO** ($a = e^{-\gamma\epsilon}$)
O: $\mathbf{m}_{t+\frac{1}{4}} = \sqrt{a}\,\mathbf{m}_t + \sqrt{(1-a)TM}w_t$
B: $\mathbf{m}_{t+\frac{1}{2}} = \mathbf{m}_{t+\frac{1}{4}} - \frac{\epsilon}{2}\nabla_{\boldsymbol{\theta}}\tilde{U}(\boldsymbol{\theta}_t)$
A: $\boldsymbol{\theta}_{t+1} = \boldsymbol{\theta}_t + \epsilon M^{-1}\mathbf{m}_{t+\frac{1}{2}}$
B: $\mathbf{m}_{t+\frac{3}{4}} = \mathbf{m}_{t+\frac{1}{2}} - \frac{\epsilon}{2}\nabla_{\boldsymbol{\theta}}\tilde{U}(\boldsymbol{\theta}_{t+1})$
O: $\mathbf{m}_{t+1} = \sqrt{a}\,\mathbf{m}_{t+\frac{3}{4}} + \sqrt{(1-a)TM}w'_t$

that covers typical stochastic gradient Langevin dynamics[25] and stochastic gradient Hamiltonian Monte Carlo[26]. With $T = 0$, equation (3) becomes the time continuous version of stochastic gradient descent with momentum[23]. This similarity allows to reuse prior knowledge on damping $\gamma$ or the discrete step-size $\epsilon$ from classical deep learning trainings. The advantages of Hamiltonian Monte Carlo mainly depend on its Metropolis-Hastings acceptance step[27]. This eliminates a bias caused by using the proposal points computed with the dynamical system as is. To use the advantages of acceptance steps in stochastic gradient sampling, we consider different implementations of the discrete dynamics of (3). Therefore, three methods are chosen and Table 1 gives an overview. For other discretizations we refer to Leimkuhler et al.[28] for a full overview. The algorithms differ in symmetry[24,29,30] as it is required for acceptance steps. The only method to apply acceptance steps in the stochastic gradient setting was AMAGOLD proposed in ref. 29. This paper approaches to overcome the convergence problems on deep learning tasks with a new Metropolis-Hastings acceptance proposal. These acceptance steps stabilize stochastic dynamics and as they are essential for the success of Hamiltonian Monte Carlo we evaluate their success in Bayesian neural network inference. Acceptance steps are used in two methods that allow for efficient Metropolis-Hastings acceptance steps in stochastic gradient driven sampling of neural networks to make use of the robust and knowledge independent sampling qualities known from Hamiltonian Monte Carlo. Our key findings:

- We propose generalized SGHMC, a new scalable Metropolis-Hastings corrected sampling algorithm that accepts simulation steps based on batch-evaluations[21,31]. This method allows to run efficient step-wise acceptance while introducing an extrema preserving bias.
- We propose Hamiltonian Trajectory Ensemble (HTE), an ensemble method based on stochastic gradient trajectories that does not converge to the true posterior, but uses acceptance steps in an ensemble setting.
- We grid search hyperparameters for our algorihtms as well as all competitor SGHMC methods demonstrating the critical effect of the step size $\epsilon$ in any SGHMC simulation.
- We demonstrate, that the size of an Bayesian ensemble is not decisive for its accuracy, but its calibration. Therefore, we run a greedy sub-sampling experiment tried to eliminate samples from the Bayesian ensembles.

In essence, our methodology provides the full pipeline from fine-tuning the sampler to improving efficiency of test inference, demonstrating the potential of Bayesian ensembles in deep learning. Prediction accuracy and calibration scores are pushed beyond the limits of deterministic and variational Bayesian methods without the costs of classical Hamiltonian Monte Carlo.

## Results

The results section is put ahead of the details on methods 4. While our evaluation demonstrates the behavior of stochastic gradient sampling methods on deep learning problems as well as the performance of the proposed methods, it also carefully revises existing methods and showing their comparison. All methods are chosen based on compatibility with deep learning. Further discussion follows in section " Previous acceptance step approaches". We start with an artificial sampling experiment from a multivariate normal, comparing the two methods proposed in this paper with the two direct competitors.

### Synthetic benchmark

To create a neural network related benchmark for stochastic gradient sampling algorithms, we sample multivariate Gaussian distribution with diagonal covariance matrix. The algorithms run on the exact

**Table 2 | Phases of the hyperparameter study**

| Phase | Parameters | Tested influences |
|---|---|---|
| **(1)** | std. of $p(\boldsymbol{\theta})$ / weight decay | step size |
| | {200, 20, 5, 1, 0.02, 0.002} | {30, 10, 3, 1} $\cdot$ $10^{-5}$ |
| **(2)** | RMSprop | step size |
| | {yes, no} | {3, 1} $\times$ {$10^{-5}$, $10^{-4}$, $10^{-3}$} |
| **(3)** | cosine schedule | final step size & RMSprop |
| | {yes, no} | {$10^{-5}$, $10^{-6}$} $\times$ {yes, no} |
| **(4)** | momentum | step size |
| | {0.9, 0.99} | {10, 3, 1} $\cdot$ $10^{-5}$ |
| **(5)** | batch size | step size |
| | {80, 400} | {10, 3, 1} $\cdot$ $10^{-5}$ |
| **(6)** | integrators and acceptance | cp. Section "Ablation Study" |

The variations for each parameter is given a number, which is referenced in the following.

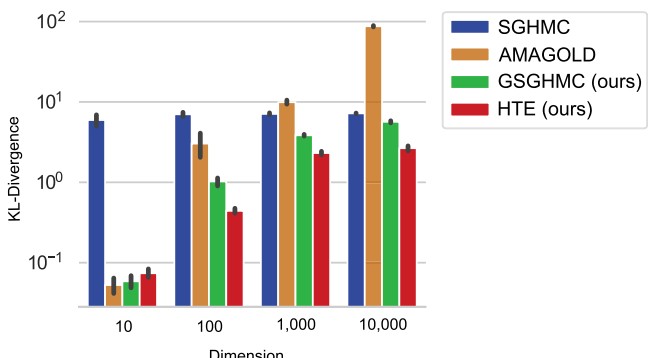

**Fig. 1 | Gaussian testing.** Benchmarking the sampling algorithms on exploring independent Gaussian distributions (10 runs) with 1000 samples while increasing the dimension of the test distribution. Error bars show the standard deviation.

gradients of the negative log-likelihood disturbed by standard normally distributed noise $n \sim N(0, 1)$. For the set of Gaussian distributions we sample the means from an exponential distribution Exp(0.1) and the variances from a uniform distribution Unif[0.1, 1]. This is close to the expected mean-field model in many variational approaches on Bayesian neural networks[6]. Similar to the following experiments on image datasets, the number of samples was limited to 1000 and the tuning was done by structural variations of the step size. The results are shown in Fig. 1. We observe increasing Kullback-Leibler divergence between the true and estimated distribution, with increasing number of dimensions. While AMAGOLD outperforms all competitors at dimension 10, the other acceptance methods start to outperform AMAGOLD and SGHMC from dimension 100 on. From dimension 1000 the Kullback-Leibler divergence stays constant for the proposed algorithms.

## Exploration of trajectory parameters

The experiments on image datasets focus on two different neural network architectures, which cover aspects of modern deep learning architectures, but remain reasonable small to allow ablation study experiments. A total number of 50,000 epochs per model for the first experiment only made this necessary. We divide the ablation experiments in phases showing the considered parameter dependencies. An overview is given in Table 2. The first model is a LeNet[32] architecture. It represents purely feed-forward architectures based on a feature extractor part with convolutional layers and a following dense classifier. Although larger architectures from this group of models can outperform the small models, its size allows us to compute the large-scale experiments in a reasonable amount of time. Since the effect of hyperparameters in deep learning tasks can be crutial for performance, we present this study for stochastic gradient Hamiltonian Monte Carlo. The costs made us choose the second model too, which represents models with residual skip connections. We use a Wide-ResNet-10-2, which we simply refer to as ResNet throughout this section,[33]. While the LeNet model has about 270,000 parameters, the ResNet has about 300,000. We train the LeNet model on the balanced EMNIST dataset[34] including 47 classes of MNIST-like images of different characters. The ResNet is trained on the very popular CIFAR-10 dataset[35]. As in Izmailov et al.[36], the batch normalization layers are replaced with feature response normalization to keep the samples independent of each other[37]. In both cases, the default train-test split is 95%/5% training/validation data leaving 107,160 train images for EMNIST and 47,500 for CIFAR-10.

In absence of the true predictive distributions, uncertainty quantification is usually compared on performance and calibration metrics, which allow the comparison of the predicted mean and therefore with deterministic neural networks as well. We choose

accuracy metric for the performance and expected calibration error (ECE) for calibration assessment as the most common ones for balanced classification tasks. The metrics are all computed based on test sets using the datasets default train/test split (CIFAR-10: 10,000; EMNIST: 18,800). Section "Bayesian neural network comparison" contains further comparison with other Bayesian inference algorithms applied to these neural networks.

With the similarity of the Euler-Maruyama discretization (cp. Eq. (3)) to stochastic gradient descent (SGD), we apply prior knowledge on momentum decay $\alpha$, which is usually chosen from [0.9, 1.0]. We restrict the choice of $\epsilon$ and $\gamma$ to satisfy the constraint

$$(1 - \epsilon\gamma) = \alpha. \qquad (5)$$

For the equations in Table 1 only one free parameter remains: The step size $\epsilon$. For very small step sizes we propose a cosine schedule as it was referenced in Table 2. Following the constraint, the step size schedule keeps the momentum variable constant and adapts both step size $\epsilon$ and damping parameter $\gamma$ accordingly to increase $\epsilon$. This method speeds up the convergence of burn-in, before the sampling phase starts.

While we concentrate on the algorithmic parameters from phases (2) on, we start with the weight decay parameter and get an overview on loss training curves, when we apply the Langevin dynamics (cp. Eq. (3)). The weight decay regularization appears similar to the regularization through the prior in the Gaussian case differing only in a scaling factor of the training set size. In the context of deep learning we will use the term weight decay from here on. In Fig. 2 we show an subset of the generated trajectories. For different step sizes, we observe the following influence of varying weight decay: Up to weight decays 2.0 and 0.02 for the LeNet and ResNet respectively, the trajectories are very alike. Above this threshold the loss remains considerably larger and we observe occasional instabilities as in the upper left corner (cp. Fig. 2). We choose the weight decay to be at least above the stability threshold to be able to converge to useful minima for the given classification problem. Since weight decay gives considerable performance gains on deterministic methods, we argue to not choose it too small either. In the neural network architectures analyzed in this paper, the instabilities observed with weight decays did not occur in the deterministic case, what sheds light on the importance choosing this parameter for the sampling process. During our experiments we shared the results of ref. 36 that cold posteriors[23] are not necessary to get competitive results. Considering $T$ as a parameter, we fix it to $T = 1$ throughout the following experiments (cp. Eq. (3)).

For the following simulations we run 5 parallel sampling chains with a burn-in of 1000 epochs and collect 1000 models during

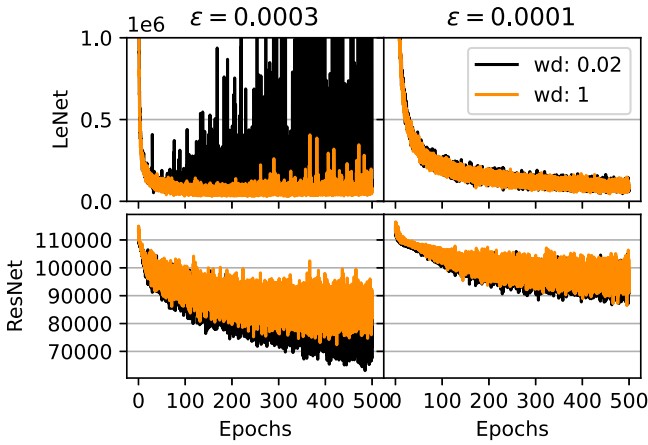

**Fig. 2 | Hyperparameter exploration, Phase (1).** This figure shows a selection of trajectories under different weight decays wd (standard deviation of $p(\boldsymbol{\theta})$) and step sizes $\epsilon$. The training progress visualizes the expected problems and preferred burn-in trend of the following simulations.

sampling phase after 10 epochs each, starting at a random network initialization.

Figure 3 shows the results of phases (2) and (3). The figure shows the performance measures over the step size, while the momentum variable is fixed to 0.9. The red and blue lines show the results for baseline SGHMC and SGHMC including a curvature estimate similar to RMSprop[38]. The two models are separated by the different line-types. We set the range of step sizes from small step sizes that show progress during burn-in until large step sizes, which caused divergence of the trajectory. Along the baseline methods, we compare both methods with our constant momentum learning rate schedule, observing top calibration scores in these simulations. For the accuracy metric, the baseline methods scores slightly better on the LeNet problem (0.1%) and a lot better on the ResNet experiment (2.8%). Most importantly, the plot shows the behavior of accuracy scores with increasing step size. In multiple cases, the top performance is achieved with the largest step size close to the stability threshold. In all cases, the step above the last shown point to increasing step size caused the dynamics to diverge, what we represented with nan. In most cases, the top performance in accuracy was also the best calibrated Bayesian ensemble.

The results of Phases (4/5) have been moved to Appendix A. We ran the top experiment from the step size variations (cp. Fig. 3) checking whether it remains the top simulation in both curvature estimation scenarios. First, we changed the momentum value to 0.99 testing its influence on the optimized sampling. Second, in the last column the batch size is changed from 80 to 400. We observe small improvements towards the larger batch size, but the changes are negligible in comparison to the effect of changing the step size of the simulation. Still, for the following simulations, we keep the larger batch size.

The experiments so far demonstrate parameter dependencies in stochastic Langevin-dynamics applied to typical deep learning tasks. The following experiments include different integrators, comparing all main methods applicable on deep learning tasks. This includes the comparison of acceptance steps methods for more accurate Markov Chain Monte Carlo trajectories.

## Ablation study

To understand the effects of the different components of the algorithms used throughout this paper, this section summarizes results on SGHMC chains with different integrators as well as different acceptance step algorithms. We observe multiple interesting results shown in Table 3. First, we observe the AMAGOLD algorithm to perform

poorly regarding accuracy. The need to get an acceptance rate high enough to finish the simulation occasionally limits the possible step-size range. While we also tried to speed up the burn-in using our initial learning rate schedule, we report on the best results we achieved. Second, our proposed method for OBABO acceptance performs similar to its non-acceptance counterpart. While it considerably improves over AMAGOLD acceptance simulations, the question arises how much influence the unbiased trajectory has on the ensemble performance. The following section evaluates the difference by repeating simulations with different random seeds, but keeping the simulation parameters fixed. The effects of the stochastic proposal algorithm using the Hamiltonian acceptance step, results in performance boosts on EMNIST and CIFAR-10 classification. For the LeNet on EMNIST it reaches a plus of 0.7%, for the ResNet it reaches 5.2% over the Euler ensemble. At the same time, the calibration measures decreases in comparison to the best compatitors, but remains in the range of the Euler simulations.

As we already discussed, the acceptance rate was one crutial factor to influence the performance metrics reported here. While there is often an optimal acceptance rate to balance exploration and exploitation, we faced some issues deciding on hyperparameters with the acceptance rate. Mostly, because the changes in step-size, that were required to change the acceptance rate where much larger than the changes we observed to change the accuracy by multiple percentage points. We decided to leave the acceptance rate aside for the tuning phase, although it was still important to finish simulations. The typical dynamics of the acceptance rate was starting at almost 100% at the beginning of burn-in and settling at a stable value for the rest of the simulation. We ended up with acceptance rates of about 50% for both our algorithms with the most promising hyperparameter configuration regarding the evaluation metrics. While this important quality of the sampling algorithms is mainly left aside for the proposed evaluation, we extend the discussion in Appendix C.

## Bayesian neural network comparison

This section compares the results on sampling algorithms in this paper to different Bayesian neural network approaches. In particular, we compare against regular deep ensembles and optimization based variational inference approaches as well as deterministic trainings. We include two different methods for variational inference on Bayesian neural networks: The Bayes by backpropagation approach[2,39] extended by the flipout method[40] and the variational online Gauss Newton (VOGN) method proposed in ref. 41 and proven to be competitive in ref. 42. We compare based on accuracy and expected calibration error again, but also report maximum calibration error and the number of epochs to account for the computational costs independent of the hardware setup. For an improved overview on the field MC-dropout, stochastic weight averaged Gaussian (SWAG) and Cyclical learning rate schedulers are included[10,12,43]. In this section, we repeat the simulations 5 times with different random seeds to report on statistical variation, which was not possible for all ablation experiments.

In addition to the hyperparameter tuning study, we add a regression experiment of timeseries forcasting for this final comparison. We experiment with the chaotic Lorenz'96 differential equation system, which is a common benchmark for timeseries forcasting problems. The system is simulated with 5 states, of which 50 steps are used for intializing the model and 50 are predicted for computing the loss. The evaluation predicts 150 timesteps ahead, on which we compare maximum absolute error (MAE), mean squared error (MSE) and the $R^2$ score. The model is a one layer LSTM with ReLU activation with a state-space of 100. Although this problem was not included in the hyperparameter study, we observe similar results, what demonstrates the generality of the results. Comparing the results in Table 4, we see the top results are achieved with the Bayesian sampling algorithms. Still, we observe differences between the problems: While the ensemble

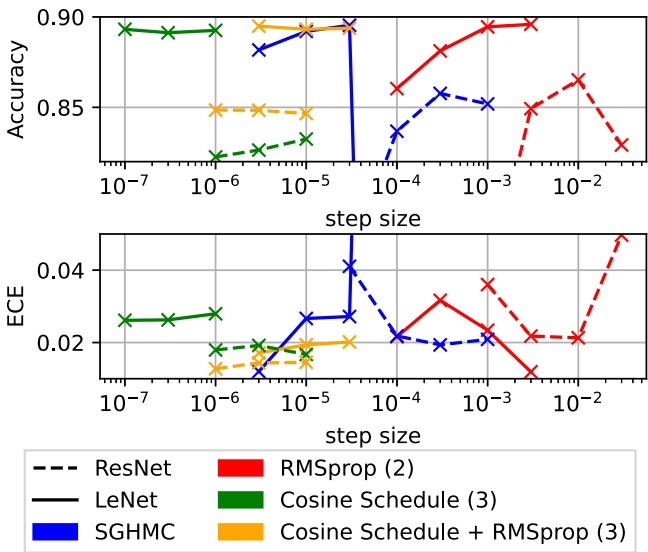

**Fig. 3 | Hyperparameter exploration, Phase (2/3).** Studying the dependence of accuracy and calibration on the sampling step size while running Euler discretization. We also evaluate the usage of curvature estimates with varying step size and a new step size schedule approach to shorten the burn-in with small step-sizes.

created from different model initializations does consistently outperform the deterministic baseline, it sets the strongest baseline in the timeseries forcasting case. The HTE algorithm does only match this baseline, while the other Bayesian inference algorithms could not reach this performance. Also, the two classification problems appear to be quite different. While the EMNIST classification benefits from all sampling algorithms equally, the CIFAR-10 classification mainly benefits from the HTE mechanism. We approach an explanation in the discussion. The results confirm a performance increase of Markov Chain Monte Carlo algorithms over variational inference based methods on the considered problems including the Bayesian neural networks. The costs for this performance remains the increased computational burden in comparison to variational inference methods. Although the proposed sampling methods considerably reduce the costs in comparison to Hamiltonian Monte Carlo experiments as in ref. 36, the number of epochs against the deterministic training is increased considerably, as we show in the column reporting the number of epochs these algorithms ran on average. These costs need to be balanced against the performance and calibration increase from Table 4.

We also report on out of distribution detection on two additional datasets. For the EMNIST classification we compute the predictive entropy for the train dataset of Fashion MNIST[44]. For the CIFAR-10 classification we use the test split of the Street View House Numbers[45] dataset. The results are visualized in Fig. 4 by showing the sorted entropy scores over the out of distribution dataset. The shaded area shows the standard deviation over the five random seed repetitions from Table 4. The optimal curve would be returning the maximum entropy for all datapoints, so the higher the curve, the better the out-of-distribution detection. We observe HTE to outperform its competitors, although not being comparably calibrated. The GSGHMC curve is at least partly above the SGHMC baseline, but does not meet the HTE line.

**Ensemble size reduction**

The ensembles of 1000 members make the evaluation and prediction process considerably more costly than running the inference of a single model. That raises the question how many models need to be kept to maintain the performance achieved with the large ensemble, but reducing the cost of evaluation. Therefore, we run a simple

greedy selection on the ensembles, keeping only samples that appear to be essential for a given metric not to be impaired. We visualize the results in Fig. 5. The number of remaining ensemble members is displayed on the y-axis after the reduction of ensembles from the prior section. The variance is computed both over the different random seeds and 5 greedy repetitions with different ordering each. We can see that the Accuracy and Calibration have quite different requirements with regards to the ensemble size. While preserving the accuracy requires less than 50% of the ensemble, the calibration requires almost all of it. While there are methods to compress Bayesian ensembles in a principled manner[46], we wanted to demonstrate, that the Bayesian sampling is not necessarily the last step before having a model ready for production. The advances in accuracy are less costly to preserve than the diversity needed for calibration. That applying a reduction technique can still be favorable is further discussed in Appendix F.

## Discussion

In this study, we demonstrate the effective application of Metropolis-Hastings acceptance steps in stochastic gradient Hamiltonian Monte Carlo for image classification using deep learning models. The proposed methods achieve stable sampling chains, enabling the construction of ensembles that exhibit superior prediction performance compared to deterministic and variational approaches.

Unlike prior work on Metropolis-Hastings in large dataset settings[43,47–51], our methods support an exploratory parameterization of the sampling chain with an fixed computational budget. Notably, on both datasets, this parameterization leads to substantial accuracy improvements, attributed to the critical relationship between step size and predictive performance.

Our results systematically compare previously reported sampling algorithms with a comprehensive exploration of parameter settings, ensuring a fair evaluation framework that, to our knowledge, has not been addressed in the existing literature. The findings align with trends observed in Hamiltonian Monte Carlo methods applied to CIFAR-10 but achieve computational feasibility for a broader research community. For instance, Izmailov et al.[36] required a 512-GPU cluster, whereas our implementation ran efficiently on a single GPU. While the computational costs of sampling methods increase over deterministic and variational methods, this study demonstrates their potential in non-convex optimization.

Overall, this work compares a range of posterior approximation methods for Bayesian neural networks, establishing a robust baseline that facilitates direct comparisons and advances research in this domain.

Hyperparameter exploration was the initial focus of our experiments. Consistent with findings on deterministic neural networks, the choice of learning rate and regularization parameters plays a critical role in the performance and generalization of trained models[52,53]. We extend this observation to the stochastic gradient sampling case, noting a strong dependence of model accuracy on the step size used in trajectory simulations across all experiments. We stress again, that there was no relation between acceptance rates and the actual performance, which we mainly analyzed in terms of accuracy. This is further discussed in Appendix C.

A common hypothesis regarding the quality of minima associated with good generalization properties relates to their extend. Large learning rates often lead optimization algorithms to converge to wide and large minima, which are thought to promote better generalization. We propose that this principle may also apply to sampling algorithms, where larger step sizes might bias the sampling chain toward wide minima, potentially enhancing the generalization of the resulting ensemble. Our experiments aimed on taking this advantage to the Bayesian ensembles, potentially creating additional divergence to an optimal sampling regarding the distribution estimation.

**Table 3 | Phase (6): Studying different integrators and acceptance steps**

| | Integrator | Acceptance | Accuracy | NLL | ECE |
|---|---|---|---|---|---|
| LeNet/ EMNIST | Euler | no | 89.4% | 26.95 | 2.3% |
| | Skew-sym. | no | 89.8% | 22.89 | **1.0%** |
| | OBABO | no | 90.0% | 22.90 | **1.0%** |
| | Skew-sym. | AMAGOLD | 88.6% | 27.93 | 2.4% |
| | OBABO | GSGHMC | 90.0% | 22.83 | 1.1% |
| | OBABO | HTE | **90.1%** | 22.07 | 2.7% |
| ResNet/ Cifar-10 | Euler | no | 86.5% | 33.49 | 2.1% |
| | Skew-sym. | no | 86.6% | 31.36 | **1.5%** |
| | OBABO | no | 86.9% | 30.99 | 1.7% |
| | Skew-sym. | AMAGOLD | 83.7% | 37.35 | 2.2% |
| | OBABO | GSGHMC | 86.8% | 39.11 | **1.5%** |
| | OBABO | HTE | **91.7%** | 27.35 | 2.4% |

Each result is the maximum of a step-size variation around the Euler optimum as in Figure 3. Best values are marked as bold.

Another approach to explain the importance of exploration with large learning rates can be driven by loss-landscape visualizations[54]. As we show in the appendix the two problems we analyzed represent very different local structure. (cp. Supplementary Fig. D3). While the ResNet has many close minima, on the LeNet problem only one minimum is visible in the chosen range. This could be a reason for the smaller effects with large learning rates on the LeNet model as well as the effects, when using the cosine step-size schedule. The cosine schedule experiments naturally focused on local exploration and falls short on the experiments without the schedule in terms of accuracy (cp. Fig. 3). The results on the timeseries prediction task support the hypothesis, that ensembles on timeseries models also benefit from exploration of different minima, which would be similar to the ResNet case apparant from Table 4.

The results highlighting the critical role of step-size selection reveal a practical challenge: identifying optimal step sizes for sampling chains is computationally expensive. The choice on the step-size can be simplified with analytical observations in the convex-case[55–57]: For Hamiltonian Monte Carlo (HMC), a common heuristic involves setting the step size to the smallest expected standard deviation[21] for example. However, the absence of interpretability for parameters in black-box models complicates the application of these results and heuristics. Furthermore, the non-convex nature of deep learning problems raises fundamental questions about the meaning of standard deviation in the posterior distribution. Comparing the results on the sampling chains to deterministic and variational deep learning, we observe a similar role of tuning the step-size. This was not expected, since the sampling should average out the need to find the exact minimum.

For non-convex problems, convergence analyses such as those in ref. 58 impose a constraint restricting sampling to the region within a ball. This approach encounters limitations in the context of deep neural networks due to the lack of prior knowledge about parameter range. Advancing research in this area could enable to focus computational effort to regions of the likelihood landscape that are more promising, as opposed to the current state-of-the-art approach, which relies on uninformed exploration. Currently, the exploration of multiple minima was found to improve prediction results[14], but the interaction with the data and model complexity remains unknown. Our convergence analysis on a small benchmark problem showed rapid convergence for medium and large step sizes and multi modal distributions (cp. Appendix G). In this analysis, the proposed GSGHMC and HTE methods show improved mixing over their competitors.

Our ensemble reduction experiment represents an initial step toward addressing this challenge. As anticipated, we observed an over-representation of redundant information within ensembles comprising hundreds of models, enabling significant reductions in ensemble size without substantial performance loss regarding accuracy. In the randomly sampled stochastic gradient Hamiltonian Monte Carlo ensembles, we hypothesize that the extraction of informative features is dominated by the most descriptive and widely relevant features. However, a considerable proportion of noisy features persists, hindering further reductions in ensemble size. The calibration benefit however, is not preserved, when we reduce the ensemble in an uninformed manner. This could be caused by the slightly varying features that cause the diversity contributing to improved calibration, as evidenced by comparisons of the expected calibration error. The behavior of the Hamiltonian trajectory ensembles suggests similarities to pure Hamiltonian Monte Carlo, which converges effectively to minima between momentum resampling steps and the log-likelihood as well as parametrization can positively and negativly influence the behavior of the total ensemble.

We were surprised by the minimal impact of the RMSprop curvature estimate[59] and variations in integrators observed during our experiments. While the Euler discretization led to a slight reduction in calibration, the OBABO and skew-symmetric integrators did not exhibit a significant independent effect. We suggest, the curvature estimate might be less effective in an optimization driven by gradients and independent noise causing no increase in exploration and barely effecting the influence of the step-size.

In a broader context, the hyperparameter search highlights the necessity of exploring the parameter space and exposes costs and problems during this search. We explicitly refer to step-size regions close to the divergence barrier. If we facilitate exploration, we find improvements in accuracy and calibration for the problems analyzed in this paper. However, the prediction performance depends on the hyperparameters such that the algorithms do not unfold their potential when choosing small step-sizes. These results are achieved with models comparably small for the deep learning community. The behavior on larger models is therefore hard to predict. At first, small models on complex datasets seem to benefit a lot from the diversity in large Bayesian ensembles. On the contrary, large models in relation to the complexity of the dataset have been observed to benefit not quite as much in variational settings[39]. As each model can only predict based on a fixed weighted selection of visual features, we argue that diversity in these features through multiple models should help to improve the robustness in large models also. With our sampling methods, these models can be part of different minima and therefore include new aspects for classification tasks. What remains an open question is whether there are inference methods, which guarantee the required diversity. While the HTE ensemble considerably improved the accuracy, it could not improve the calibration much. The generalized stochastic gradient Hamiltonian Monte Carlo method on the other hand could not equally improve accuracy. An approach to combine the two methods is not possible as is, but might be topic for future research.

Although this paper extensively searched through relevant parameters, some simplifications have been necessary that could not be included. One was the choice of the prior. Choosing a meaningful prior is difficult in the black-box setting of deep learning. Although different studies pointed out advantageous prior choices using other distributions than the typical Gaussian,[60,61], these have not yet been transferred to variational approaches. Heavy tails on the distribution are standing out as a beneficial choice for the sampling algorithms evaluated. Since the Gaussian assumption allows to connect the prior to an $L_2$ regularization effect as well as keeps the comparison to variational methods fair, we decided to keep this constraint. Another limitation was our constraint on the relation between friction and step size of the discretization. Although other papers applied similar simplifications[26],

**Table 4 | Comparison of Bayesian neural network approaches showing the mean and standard deviation of 5 seeded repetitions**

| | | Method | Accuracy ↑ | ECE ↓ | MCE ↓ | Epochs |
|---|---|---|---|---|---|---|
| LeNet | EMNIST | deterministic | 85.6% ± 0.7% | 2.0% ± 0.4% | 17.5% ± 6.5% | 100 |
| | | MC dropout | 87.3% ± 0.2% | 3.5% ± 0.3% | 24.6% ± 8.8% | 100 |
| | | SWAG | 87.1% ± 0.2% | 3.7% ± 0.5% | 37.6% ± 25.3% | 100 |
| | | variational inference | 87.4% ± 1.2% | 2.0% ± 0.5% | 17.3% ± 2.2% | 500 |
| | | VOGN | 86.0% ± 1.4% | 1.7% ± 0.4% | 16.2% ± 2.4% | 500 |
| | | ensemble | 88.9% ± 0.1% | 1.1% ± 0.1% | 33.2% ± 31.8% | 3200 |
| | | SGHMC | **90.0% ± 0.05%** | 1.0% ± 0.1% | 31.3% ± 28.1% | 10000 |
| | | Cyclical Cosine | **90.1% ± 0.08%** | 3.3% ± 0.04% | 24.9% ± 2.7% | 5000 |
| | | GSGHMC (ours) | 89.5% ± 0.05% | **0.8% ± 0.02%** | **14.4% ± 4.5%** | 5000 |
| | | HTE (ours) | **90.0% ± 0.1%** | 0.9% ± 0.03% | 23.2% ± 9.4% | 5400 |
| ResNet | Cifar-10 | deterministic | 86.4% ± 0.6% | 5.8% ± 1.8% | 22.9% ± 5.2% | 100 |
| | | MC dropout | 86.7% ± 0.3% | 4.6% ± 0.2% | 23.2% ± 4.4% | 100 |
| | | SWAG | 86.4% ± 0.4% | 4.5% ± 0.6% | 24.2% ± 3.2% | 100 |
| | | variational inference | 87.0% ± 0.2% | 2.1% ± 0.4% | 12.2% ± 7.2% | 500 |
| | | VOGN | 86.0% ± 0.4% | 2.3% ± 0.4% | 20.3% ± 9.0% | 500 |
| | | ensemble | 87.9% ± 0.4% | 2.3% ± 0.4% | **9.5% ± 3.5%** | 3200 |
| | | SGHMC | 86.8% ± 0.05% | 1.5% ± 0.1% | 11.5% ± 5.5% | 10000 |
| | | Cyclical Cosine | 92.0% ± 0.1% | 4.7% ± 0.1% | 26.6% ± 5.7% | 10000 |
| | | GSGHMC (ours) | 86.8% ± 0.2% | **1.4% ± 0.1%** | 9.6% ± 5.3% | 10000 |
| | | HTE (ours) | **92.2% ± 0.1%** | 2.3% ± 0.1% | 22.9% ± 6.3% | 2000 |
| | | | MAE ↓ | MSE ↓ | $R^2$ ↑ | Epochs |
| LSTM | Lorenz'96 | deterministic | 0.216 ± 0.011 | 0.104 ± 0.011 | 0.454 ± 0.061 | 800 |
| | | MC dropout | 0.191 ± 0.003 | 0.076 ± 0.002 | 0.602 ± 0.013 | 800 |
| | | SWAG | 0.211 ± 0.006 | 0.098 ± 0.011 | 0.487 ± 0.056 | 800 |
| | | variational inference | 0.214 ± 0.0004 | 0.077 ± 0.0003 | 0.596 ± 0.001 | 3200 |
| | | ensemble | **0.182 ± 0.001** | **0.068 ± 0.001** | 0.643 ± 0.006 | 16000 |
| | | SGHMC | 0.193 ± 0.002 | 0.078 ± 0.002 | 0.591 ± 0.008 | 20000 |
| | | Cyclical Cosine | 0.197 ± 0.002 | 0.084 ± 0.002 | 0.561 ± 0.011 | 20000 |
| | | GSGHMC (ours) | 0.185 ± 0.002 | 0.075 ± 0.001 | 0.608 ± 0.007 | 20000 |
| | | HTE (ours) | **0.182 ± 0.001** | **0.068 ± 0.001** | **0.644 ± 0.0004** | 11000 |

We compare model performance by Accuracy, model calibration by expected and maximum calibration error (ECE/MCE) and computational costs by the total number of epochs for one experimental run. Bold values indicate the best results for each metric.

an increased variability of simulation parameters could show even further improved results or stabilize the dynamics with varying hyperparameters. Using knowledge from deterministic trainings of deep neural networks remains a time and cost efficient way of setting a hyperparameter prior and finally allows to compare the ensembles to deterministic trainings and use them for our understanding of deep learning in total. The extend of these simplification remains an active field of research. Furthermore, the number of samples was chosen arbitrarily to 1000 in most cases. To evaluate the effects of considerably larger and smaller sampling size can hint on the effects of knowing a single minimum very precise versus a reduced training cost. Further applications of acceptance rates might be automatic parameter tuning algorithms such as NUTS[18] or methods to distinguish aleatoric and epistemic uncertainty influences[62] for active learning tasks. A thorough understanding of the mechanisms is the key to know, when the additional cost of sampling methods can give additional performance benefits.

We presented two efficient sampling algorithms for the inference of Bayesian neural networks. Based on an extensive hyperparameter study, we demonstrated the performance benefits achievable with both methods. The main advantages distribute between the methods. GSGHMC emerged as the top-performing algorithm in terms of calibration among SGHMC-based approaches, HTE achieved the largest accuracy gain. Although HTE does not converge to the true posterior,

it allowed to create an algorithm with strong mixing properties focusing on exploration of the unknown posterior. Both algorithms achieved significant performance improvements, with benefits of 4.3% and 6.0% accuracy over state-of-the-art Bayesian sampling methods. These results raise intriguing questions about the interplay between feature variation across local minima and the model properties that contribute to improved calibration. Further research in this direction could deepen our understanding of how to optimize Bayesian neural networks for both accuracy and reliability.

## Methods

To introduce the concepts of Metropolis-Hastings acceptance we review the key concepts in the following and introduce the background to the two sampling algorithms, which have been evaluated in this paper. Afterwards, we introduce the generalized stochastic gradient Hamiltonian Monte Carlo algorithm and the Hamiltonian trajectory ensemble.

### Errors in Markov Chain Monte Carlo estimation

There are multiple relevant errors that occur during the numeric approximation of the posterior probability distribution with Markov Chain Monte Carlo[28]. However, a moment matching at time step $t$ does not exactly gather the error the user of a sampling method is interested in. For Markov Chains the long term error is most relevant, to compute

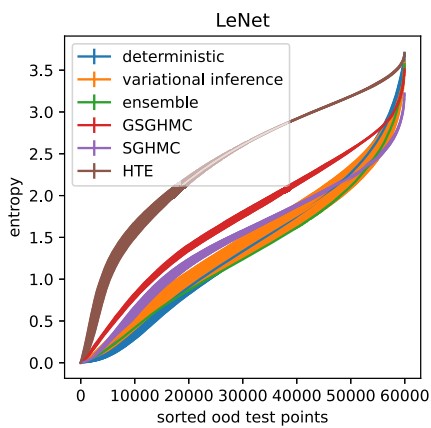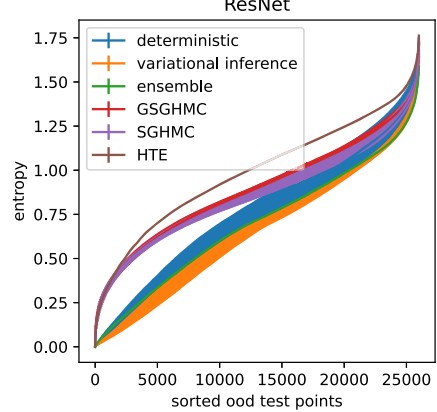

**Fig. 4 | Entropy-based out of distribution detection.** Comparison of out of distribution detection via predictive entropy. The LeNet example uses Fashion MNIST and the CIFAR-10 example the Stanford Street View House Numbers dataset, which both don't share any intersection with the training or testing data. The higher the curve, the better. The shaded area shows the variance over the 5 repetitions from Table 4.

correct Monte Carlo expectations. Therefore, the representation of $p(\boldsymbol{\theta}|S)$ through the ensemble is what matters. It consists of an error caused by discretization (cp. Eq. (3)), which is proportional to the step size. Also, there is an error due to the limited number of samples collected during sampling phase. These errors are often referred to as sampling bias and sampling error, respectively. The error or bias caused by the discretization can not be reduced by running the simulation for a longer time. On the other hand, a reduction of the step size reduces the mixing of stored Markov Chain Monte Carlo samples. Similar to a bias-variance trade-off, these two errors balance each other. Simulations with a constant number of steps do either precisely simulate the dynamics or gather more samples from a larger parameter space to represent the posterior accurately.

**Previous acceptance step approaches**

The sampling bias of Markov Chain Monte Carlo simulation can be reduced by considering the end points of trajectories proposal points $z_p$, which are accepted or rejected in a Metropolis-Hastings manner[27]. We introduce $z$ as an auxiliary state, which summarizes the concatenation of $\boldsymbol{\theta}$ and $m$, in case the sampler utilizes a momentum state. The transition probability from initial value $z_0 = (\boldsymbol{\theta}_0, m_0)$ to $z_p = (\boldsymbol{\theta}_p, m_p)$ is required to fulfill $t(z_0 \rightarrow z_p) = t(z_p \rightarrow z_0)$ for an unique stationary distribution, which is called detailed balance condition. The Metropolis-Hastings acceptance probability takes the form

$$r(z_0, z_p) = \min\left\{1, \frac{\pi(z_p)}{\pi(z_0)}\frac{t(z_p \rightarrow z_0)}{t(z_0 \rightarrow z_p)}\right\}. \quad (6)$$

We observe that non reversible Markov Chain Monte Carlo chains directly cause acceptance rates of zero. This applies in particular to Euler-Maruyama[29]. If the backward transition probability is not zero, the acceptance step guarantees reversibility by design, if the momentum is completely resampled. An advantage of acceptance steps is the stabilizing effect on the dynamics. Since stochastic gradients can prevent convergence for integrators of any order[63], the application of acceptance steps allows to run more robust sampling chains. In the following, we present the direct competitor approach of stochastic gradient acceptance steps for sampling and its limitations in the context of large models. Zhang et al.[29] propose AMAGOLD, an accumulating strategy for $r(z_0, z_p)$ using the observation that we can accept proposal points based on the product of probability densities from all state transitions between $z_0$ and $z_p$. With the log-probability this becomes a sum, which we can compute step-by-step in the minibatch-setting with little extra costs. They use the skew-symmetric integrator from Table 1 and build a skew-reversible chain, which has

non-zero acceptance rates (cp. Alg. 2). The algorithm allows to explore the posterior based on stochastic gradients, but requires $U(\boldsymbol{\theta})$ for accepting the proposal state. We run the algorithm on our test cases in all configurations tested for any other method as in Table 3 with the results in Fig. 3. In combination with Bayesian neural networks we observe small acceptance rates below 10%, if the step size is kept in a reasonable range that guarantees the optimization to move forward. To get into acceptance rates of about 30% we had to reduce the trajectory length to 2 epochs and keep the step size at $10^{-4}$, which was with factor 100 considerably smaller than without the AMAGOLD acceptance steps. The results from section "Ablation study" shows the problem of efficient acceptance steps was not overcome for Bayesian neural networks. The current solution even limits prediction accuracy. While AMAGOLD was evaluated on deep neural network cases, other competitors could not be evaluated on Bayesian deep learning problems in the literature. Efficient Metropolis-Hastings acceptance steps for Markov Chain Monte Carlo methods have also been approached in refs. 47–51. While the application to Bayesian deep learning is not in the focus of these studies, some problems occur evaluating them on the highly non-convex log-likelihoods in deep learning. The most prominent problem are variable batch-sizes, which slow down the inference of Bayesian neural networks considerably. These appear in refs. 47–49,51. The work of ref. 50 even requires occasional resampling on the data, which also limits the speed of each step and cannot be implemented with a fixed computational budget. Additionally, the batch-size is a relevant regularization hyperparameter. If it is not fixed, the knowledge transfer from regular neural network trainings is quite limited. Therefore, these methods have not been tested on any Bayesian deep learning problems yet. Together with the methological limits, these methods were excluded from the study in this paper, similar to previous studies on sampling algorithms for Bayesian deep learning[36]. We approach an algorithm that fixes the required batch-size for the gradients and acceptance step fixing the computational budget and therefore being especially compatible with deep neural networks. To do so, we rather orientate on partial momentum updates in contrast to the majority of these methods, which replace an acceptance step in an momentum resampling method. Therefore, we choose to call the method generalized SGHMC in accordance with generalized HMC.

**Our proposed Metropolis-Hastings acceptance methods**

Our two approaches aim to facilitate an application perspective. The first proposed method uses the similarity between classical SGHMC and Generalized HMC to construct an unbiased SGHMC algorithm. The second method is motivated from the long trajectory length

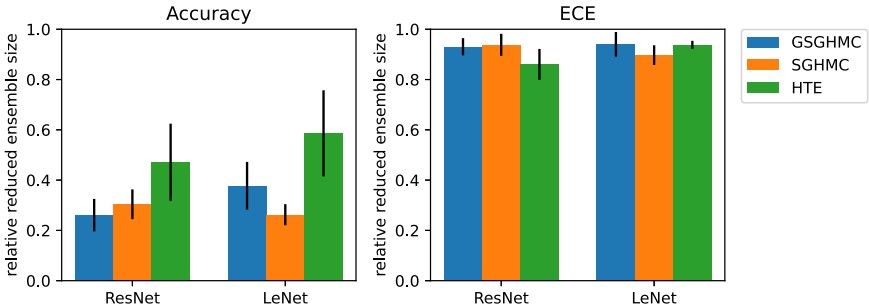

**Fig. 5 | Ensemble reduction.** Reduction of the Bayesian ensembles in a repeated greedy fashion, such that the metric in the headline is not impaired. Calibration scores require a considerably larger portion of the ensemble than accuracy scores. The error bars visualize the standard deviation over 5 repeated selections with different seeds.

parametrizations from HMC simulation experiments as in ref. 36. We ask ourselves, whether the posterior influenced by stochastic gradients might actually cause beneficial regularization, as it is well known for large models[64]. Our HMC-like method samples from this posterior and preserves the exploration properties of original HMC, while losing the exact posterior sampling property. This section presents both algorithms, which have been experimentally analyzed.

### Generalized SGHMC (GSGHMC)

The generalized SGHMC methodology uses step wise acceptance steps and is therefore composed of a batch-acceptance algorithm and an integrator, that does not result in an acceptance rate of zero. A batch-acceptance algorithm with a conservation of minima property is proposed in ref. 65. They introduce a clever notation to analyze the properties of their algorithm, which we will also use, but find the acceptance rates to be relatively small throughout their experiments. Our algorithm bases on their idea, fixing the acceptance rate by using a reversible integrator, such as OBABO. The preservation of local optima property of the algorithm is required in our case to explore the extrema of the true posterior exactly. In ref. 65 the minibatch acceptance setting is considered by an state space extension with a binary random vector $\tau \in \{0, 1\}^{|J|}$ such that $B = \{j: \tau(j) = 1, j \in J\}$. This allows to reformulate $\tilde{U}(\boldsymbol{\theta})$ to

$$\tilde{U}_\tau(\boldsymbol{\theta}) = -\frac{|S|}{|B|} \sum_{j\in J} \tau_j \log p(y_j|x_j, \boldsymbol{\theta}) - \log p(\boldsymbol{\theta}). \quad (7)$$

Since $\tau$ is independent of $\boldsymbol{\theta}$, the target distribution of the augmented system is an exponential distribution based on $\tilde{U}$ multiplied by the uniform distribution over $I_\tau = \{\tau: \sum_{j\in J}\tau_j = |B|\}$, which we denote by $\text{Unif}_B(\tau)$:

$$\tilde{\pi}(z) \propto e^{-c_n\tilde{U}_\tau(\boldsymbol{\theta})} e^{-K(\mathbf{m})} \text{Unif}_B(\tau). \quad (8)$$

The additional distribution over the momentum remains independent of $\boldsymbol{\theta}$ and $\tau$. With the target distribution $\tilde{\pi}(z)$, we can construct an Markov Chain Monte Carlo chain. Starting with equation (6), we plug in the augmented target distribution and observe that the distribution on $\tau$ does not change from initial to proposal state and does not influence the relation. The acceptance probability becomes

$$\begin{aligned}
r((z_0, \tau_0), (z_p, \tau_p)) = \\
= \min\left\{1, \frac{\tilde{\pi}(z_p)}{\tilde{\pi}(z_0)}\frac{t((z_p, \tau_p) \to (z_0, \tau_0))}{t((z_0, \tau_0) \to (z_p, \tau_p))}\right\} \\
= \min\left\{1, \frac{t(z_p \to z_0)}{t(z_0 \to z_p)}\frac{\exp(-c_n\tilde{U}_{\tau_p}(\boldsymbol{\theta}_p))\exp(-K(\mathbf{m}_p))}{\exp(-c_n\tilde{U}_{\tau_0}(\boldsymbol{\theta}_0))\exp(-K(\mathbf{m}_0))}\right\},
\end{aligned} \quad (9)$$

where each step is accepted or rejected based on a second minibatch that is selected with $\tau_p$. To reconstruct the posterior distribution $\pi(\boldsymbol{\theta})$,

we have to marginalize over $\tau$. Using the independence of the distribution on $m$, we directly compute the distribution on $\boldsymbol{\theta}$,

$$\tilde{\pi}(\boldsymbol{\theta}) \propto \pi(\boldsymbol{\theta})^{c_n} \binom{n}{|B|}^{-1} \sum_{\tau\in I_\tau} e^{c_n(-\tilde{U}_\tau(\boldsymbol{\theta})-U(\boldsymbol{\theta}))}, \quad (10)$$

which slightly differs from the distribution in ref. 65 due to our different definition of $U$. The parameter $c_n$ balances the tempering of the posterior distribution against resulting acceptance rates and overall bias. While in the proposing paper $c_n = 20$ appears as a generally good choice to trade off acceptance rate against tempering effect[65], we tried to stay at $c_n = 1$ during our simulation. By slightly increasing the step size in comparison to the SGHMC simulation without the acceptance step, our experiments ran with acceptance rates of 50.0% ± 3.5% both in the benchmark and the neural network case. We decided to keep $c_n$ fixed at this value that our algorithm results in an unbiased chain, approximating the exact posterior. The independence of $m$ guarantees the convergence results from ref. 65, which are shortly reviewed in appendix E.

The acceptance rate of a sampling algorithm often decides, whether it can be applied on large neural networks. We discuss some observations based on equation (9) that what to expect from our GSGHMC algorithm. Therefore, we take the logarithm of the acceptance probability and plug in the transition probability for OBABO as in ref. 24

$$\log\frac{t(z_p \to z_0)}{t(z_0 \to z_p)} + \log\frac{\exp(-c_n\tilde{U}_{\tau_p}(\boldsymbol{\theta}_p))\exp(-U(\mathbf{m}_p))}{\exp(-c_n\tilde{U}_{\tau_0}(\boldsymbol{\theta}_0))\exp(-K(\mathbf{m}_0))} \quad (11)$$

$$= -(K(\mathbf{m}_0) - K(\mathbf{m}_p) + K(\mathbf{m}_{3/4}) - K(\mathbf{m}_{1/4})) \quad (12)$$

$$-c_n\tilde{U}_{\tau_p}(\boldsymbol{\theta}_p) + c_n\tilde{U}_{\tau_0}(\boldsymbol{\theta}_0) - K(\mathbf{m}_p) + K(\mathbf{m}_0) \quad (13)$$

$$= c_n(\tilde{U}_{\tau_0}(\boldsymbol{\theta}_0) - \tilde{U}_{\tau_p}(\boldsymbol{\theta}_p)) - K(\mathbf{m}_{3/4}) + K(\mathbf{m}_{1/4}). \quad (14)$$

This equation demonstrates the effect of $c_n$ as it increases the acceptance rate, as long as the loss improves. The K terms depend on the gradients. We can write $K(\mathbf{m}_{3/4})$ as a function of $\mathbf{m}_{1/4}$ such that the quadratic term of $m_{1/4}$ is eliminated. Introducing $\nabla_{\boldsymbol{\theta}}\tilde{U}_{\tau_p,\tau_0}^{\text{sum}}(\boldsymbol{\theta}_p, \boldsymbol{\theta}_0) = (\nabla_{\boldsymbol{\theta}}\tilde{U}_{\tau_p}(\boldsymbol{\theta}_p) + \nabla_{\boldsymbol{\theta}}\tilde{U}_{\tau_0}(\boldsymbol{\theta}_0))$, the K-part reduces to

$$\begin{aligned}
-K(\mathbf{m}_{3/4}) \quad + K(\mathbf{m}_{1/4}) = \epsilon\,\mathbf{m}_{1/4}^\top \nabla_{\boldsymbol{\theta}}\tilde{U}_{\tau_p,\tau_0}^{\text{sum}}(\boldsymbol{\theta}_p, \boldsymbol{\theta}_0) \\
-\frac{\epsilon^2}{4}\nabla_{\boldsymbol{\theta}}\tilde{U}_{\tau_p,\tau_0}^{\text{sum}}(\boldsymbol{\theta}_p, \boldsymbol{\theta}_0)^\top \nabla_{\boldsymbol{\theta}}\tilde{U}_{\tau_p,\tau_0}^{\text{sum}}(\boldsymbol{\theta}_p, \boldsymbol{\theta}_0).
\end{aligned}$$

The first term in this equation shows as long the series of gradients agree on the direction as they are captured in $\mathbf{m}_{1/4}$, the first term is

large and dominates the second term, which is scaled by the square of the small step size. This behavior is valuable as a method to prevent U-turns in the parameter trajectory was proposed earlier in ref. 13. In case of unstable gradients and small changes in the values of U, the acceptance behavior will mostly depend on the random choice.

**Algorithm 1.** Hamiltonian Trajectory Ensemble (HTE)

> HTE($N$, U, K, $\boldsymbol{\theta}_0$, $t_L$, $\epsilon$, $M$, integrator)
> $\mathcal{S} = \{\}$ ▷Posterior samples
> $\boldsymbol{\theta} = \boldsymbol{\theta}_0$ ▷Initialize position
> **while** $|\mathcal{S}| < N$ **do**
> $\mathbf{m} \sim N(0, M)$ ▷Resample momentum
> $z = \{\boldsymbol{\theta}, \mathbf{m}\}$ ▷Store initial value
> **for** $t = 0$ to $t_L$ **do**
> integrator.step($\boldsymbol{\theta}, \mathbf{m}, \epsilon, M, \tilde{U}, T = 0$)
> **end for**
> $r = \min\{1, \exp(U(z(\boldsymbol{\theta})) - U(\boldsymbol{\theta}) - K(\mathbf{m}) + K(z(\mathbf{m})))\}$
> $v \sim \text{Unif}(0, 1)$
> **if** $r > v$ **then**
> $\mathcal{S} = \mathcal{S} \cup \{\boldsymbol{\theta}\}$ ▷Accepted!
> **else**
> $\boldsymbol{\theta} = z(\boldsymbol{\theta})$ ▷Rejected!
> **end if**
> **end while**

### Hamiltonian trajectory ensemble (HTE)

The second method we propose is motivated by the results of[36] on Hamiltonian Monte Carlo. To achieve overarching performance improvements, the Hamiltonian dynamics are simulated over a long trajectory length up to a thousand steps, using effectively gradient descent with momentum between the momentum resets, as this is what the trajectory simulation of HMC does. Although the integrators from Table 1 with stochastic gradients cannot simulate the same trajectory as the one followed by the true Hamiltonian dynamics, we know from optimization literature that the risk will reduce similarly to a gradient descent optimization (cp.[64]), which is performed when simulating the Hamiltonian dynamics in between the momentum resampling steps. Therefore, we decided to try reproducing the HMC performance benefits with an OBABO stochastic gradient trajectory, still applying the same acceptance used in HMC. While this introduces an uncontrolled bias as it was discussed in ref. 66, the divergence of the trajectories is the main concern. The aim of our method is to accept only trajectories, that stayed close to the actual Hamiltonian trajetory and are therefore excepted by the Metropolis-Hastings step. The OBABO integrator is used with $T = 0$, effectively simulating a leapfrog discretization[21] with an additional momentum decay (cp.[67]). A summary of the program realization is given in Alg. 1. We use a program-like notation, where the formulas of the integrator steps are implemented in a step method, which is applied to the variables in place for every trajectory step as given in Table 1. Evaluation of U refers to equation (2) and K to the quadratic kinetic energy term depending on $\mathbf{m}$ and $M$. The notation $z(\cdot)$ refers to the choice of $\boldsymbol{\theta}$ or $\mathbf{m}$ from $z$.

Our primary objective using this naive approach was to sample from the same likelihood, which is optimized during deep neural network training. Through the usage of stochastic gradients small minima are hidden and the likelihood gets biased towards minima that facilitate generalization[64]. Taking the acceptance probability for the GSGHMC algorithm in formula (14), the momentum terms do differ by the gradient noise. Therefore, in contrast to Hamiltonian Monte Carlo, the Kinetic energy terms do not eliminate each other. With a quadratic kinetic energy term, there is a sum of kinetic energy terms of the noise as well as scalar product terms between the momentum and the gradient noise that we explicitly ignore and that will deformate the likelihood depending on the energy in the noise. While this was argued in ref. 66 to eventually cause acceptance rates of zero, we could not empirically validate this for the gradient noise evaluated in this paper. While this does not eliminate the bias of this method, it explores a regularized log-likelihood. Applying the algorithm on neural networks, we showed to conserve benefits from regularized neural network training.

The algorithm showed surprising robustness in the neural network setting, when the dimension of the target distribution increases, a property that is observed for regular HMC[21] too. Since this method is not a Bayesian inference, but an ensemble method, we decided to use the term Hamiltonian trajectory ensemble (HTE). Advantages over regular ensemble methods from the machine learning community[68] are generated through a speed up, since it is not required to retrain the complete model in different settings. By running the trajectory length after an initial burn in creates large ensembles considerably faster than regular ensemble methods, saving time proportional to the ensemble size.

## Data availability

No data was collected for this work. The EMNIST and CIFAR-10 datasets are publicly available as well as the Fashion MNIST and SVHN datasets. The Lorenz'96 simulation is a deterministic ode solution, that can be repeated with the published code.

## Code availability

The source code for all experiments is available on CodeOcean. The Pytorch library[69] is used to construct the models and compute the gradients.

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

## Acknowledgements

Thanks to the Carl Zeiss Stiftung for funding this research under P2022-08-006.

## Author contributions

M.S. performed the theoretical analysis, the coding realization and experiments. P.M. supervised the work.

## Funding

## Competing interests

The authors declare no competing interests.
