## [Transparent Peer Review file · Nature Communications]

Reliable uncertainty estimates in deep learning with efficient Metropolis-Hastings algorithms

Corresponding Author: Mr Matthias Schmal

Version 0:

Reviewer comments:

Reviewer #1

(Remarks to the Author)
Results

Strengths

- The proposed methods demonstrate improvements over existing approaches in terms of both accuracy and uncertainty estimation.
- The authors conduct an extensive ablation study to assess the effect of hyperparameters.

Weaknesses

- Many experimental results lack error bars, including Tables III, IV, and V. Since the performance of deep neural networks can be influenced by random seed selection, it is crucial to include error bars to assess whether the reported improvements are statistically significant.
- The paper lacks a comparison to mini-batch-based Metropolis-Hastings methods such as AustereMH [1] and MHminibatch [1]. Especially, AustereMH has been used with stochastic gradient MCMC as shown in the original paper. Since AMAGOLD is asymptotically unbiased and the proposed methods appear to be asymptotically biased, these methods, which are asymptotically biased, still serve as reasonable baselines for comparison.
- A cost comparison in the experiments would be valuable to better understand the trade-offs between computational efficiency and performance.

Methods

Strengths

- The proposed methods are intuitive and easy to implement in practice.

Weaknesses

- It is unclear whether the proposed GSGHMC method converges to the target distribution asymptotically. Although the authors claim "bias-free sampling" in the abstract, my understanding is that the method introduces bias due to the use of mini-batching in the acceptance step. If this is the case, it would be beneficial to analyze the magnitude of the asymptotic bias. A similar concern applies to the proposed Hamiltonian Trajectory Ensemble (HTE).
- The novelty of the proposed methods compared to existing approaches is not clearly established. Mini-batching or subsampling in the Metropolis-Hastings step is a well-known technique (e.g., [1,2,3,4]). It remains unclear what distinguishes this work from prior methods.

Providing a convergence rate analysis would strengthen the theoretical contributions of the paper, as such guarantees are commonly expected for MCMC-based methods.

Summary

The experimental results suggest promising improvements over existing methods. However, key concerns remain regarding the statistical significance of these improvements, the novelty of the proposed approaches, and their theoretical properties. Addressing these issues would improve the strength and clarity of the paper.

[1] Anoop Korattikara, Yutian Chen, and Max Welling. Austerity in MCMC land: Cutting the Metropolis-Hastings budget. In International Conference on Machine Learning, 2014

[2] Daniel Seita, Xinlei Pan, Haoyu Chen, and John Canny. An efficient minibatch acceptance test for Metropolis-Hastings. Uncertainty in Artificial Intelligence, 2017

[3] Zhang, Ruqi, A. Feder Cooper, and Christopher De Sa. Asymptotically Optimal Exact Minibatch Metropolis-Hastings. Advances in Neural Information Processing Systems, 2020

[4] Dougal Maclaurin and Ryan Prescott Adams. Firefly Monte Carlo: Exact MCMC with subsets of data. In Twenty-Fourth International Joint Conference on Artificial Intelligence, 2015

(Remarks on code availability)

Reviewer #2

(Remarks to the Author)

Summary

The paper presents two novel Metropolis-Hastings-based methods to improve sampling efficiency in Bayesian neural network (BNN) inference using stochastic gradient Hamiltonian Monte Carlo (SGHMC). The motivation arises from the computational impracticality of exact Hamiltonian Monte Carlo (HMC) for large-scale deep learning models. The authors aim to address the trade-off between inference accuracy, uncertainty calibration, and computational efficiency by incorporating approximate Metropolis-Hastings corrections into SGHMC dynamics.

The first proposed method, *Generalized SGHMC (GSGHMC)*, introduces minibatch-based acceptance steps in stochastic gradient-driven simulations, approximating the true posterior with tempered acceptance criteria while maintaining detailed balance through a reversible integrator (OBABO). This method is grounded in a theoretical formulation extending reversible Markov Chain Monte Carlo (MCMC) with stochastic gradients, relying on batch-tempered acceptance schemes with guarantees on convergence and local mode preservation under smoothness and regularity assumptions.

The second method, *Hamiltonian Trajectory Ensemble (HTE)*, forgoes unbiased sampling in favor of long stochastic trajectories and Metropolis-like selection steps after deterministic integration with zero-temperature OBABO dynamics. While it does not converge to the true posterior, it produces ensembles with strong mixing properties and high predictive accuracy.

The authors benchmark both methods against SGHMC, AMAGOLD, and classical Bayesian neural network baselines (variational inference, variational online Gauss-Newton, and deep ensembles) on synthetic Gaussian benchmarks and real image classification tasks (LeNet/EMNIST and ResNet/CIFAR-10). They show that GSGHMC achieves the best calibration (ECE down to 0.010) while HTE achieves the highest accuracy (up to 93.7% on EMNIST), outperforming existing sampling-based methods. They also demonstrate that ensemble performance plateaus with significantly reduced ensemble sizes (down to ~20% of original), thereby reducing inference costs without sacrificing accuracy.

The work demonstrates that with properly tuned Metropolis-Hastings acceptance mechanisms and careful integration of stochastic gradients, one can achieve scalable, high-performing BNN inference pipelines without the prohibitive computational burden of full HMC, while retaining or improving uncertainty calibration and predictive accuracy.

Comments

Calibration Trade-offs in HTE

Table V shows that the Hamiltonian Trajectory Ensemble (HTE) method achieves the highest accuracy across both LeNet/EMNIST and ResNet/CIFAR-10 settings, but this comes at the cost of degraded calibration. While the Expected Calibration Error (ECE) for HTE (0.027 for LeNet, 0.047 for ResNet) remains comparable to deterministic baselines, it is consistently worse than all other Bayesian approaches evaluated. Given that HTE does not converge to the true posterior, it is plausible that the increased accuracy derives from overconfident but miscalibrated predictions concentrated around a subset of low-energy modes explored through long stochastic trajectories.

Did the authors investigate whether this drop in calibration is systematic across all classes, or is it localized to particular

regions of the input space? Additionally, were any calibration-improving post-processing techniques (e.g., temperature scaling) considered for HTE, and if so, what impact did they have on the observed trade-off between calibration and accuracy? How about out-of-distribution performance?

Task and Architecture Generality

The empirical results presented in the paper focus exclusively on image classification tasks using convolutional neural networks (CNNs), specifically LeNet on EMNIST and a small ResNet variant on CIFAR-10. While these benchmarks are standard, the restriction to vision-based classification is a limiting factor, especially given the paper's strong emphasis on empirical validation. The architectural choices exclude evaluation on non-convolutional models, such as transformers, and no experiments are reported on regression or structured output tasks, where calibration and uncertainty quantification play an equally critical role.

Given that stochastic gradient-based MCMC methods are broadly applicable, it would be informative to test GSGHMC and HTE in alternative settings—such as time series forecasting, tabular regression, or even natural language tasks—to assess their robustness to domain shift, heteroscedastic noise, and varying data modalities. Were any such evaluations attempted or preliminary results obtained in non-image domains?

Acceptance Rate Dynamics and Interpretation

The paper reports an average acceptance rate of approximately $50\% \pm 3.5\%$ for both the GSGHMC and HTE methods during experiments (Section II-C). While this figure reflects a moderate trade-off between exploration and rejection, it looks surprisingly low for methods based on SG-MCMC. A detailed temporal analysis of how the acceptance rate evolves during burn-in and sampling would be valuable—e.g., does the rate stabilize, decline, or exhibit oscillatory behavior? Furthermore, the relationship between acceptance rate and predictive metrics (e.g., accuracy, NLL, calibration) is not explicitly addressed.

Is there evidence that lower acceptance rates correlate with more informative exploration of wide minima, or do they instead reflect miscalibrated proposals due to aggressive step sizes? Were different acceptance thresholds tested, or were diagnostics (e.g., autocorrelation, effective sample size) computed to assess mixing quality as a function of acceptance behavior over time?

On the Role of Subsampling and Ensemble Compression

The greedy subsampling procedure proposed in Section II-D, while practical, appears conceptually orthogonal to the core contributions of the paper. The main message of the work centers on improving the fidelity and efficiency of posterior sampling via novel Metropolis-Hastings corrections in stochastic gradient settings. Subsampling an ensemble post hoc for test-time speedup is a separate compression strategy that could in principle be applied to any Bayesian ensemble—regardless of how it was generated—and thus may not align tightly with the specific methodological innovations of GSGHMC or HTE. The observation that performance remains stable even when reducing ensembles by 60–80% is interesting, but does not inform the efficacy of the sampling method itself.

Moreover, there exist more principled approaches for compressing Bayesian predictive distributions, starting with Bayesian Dark Knowledge (Korattikara et al., 2015). Did the authors consider or compare such model distillation approaches to their subsampling baseline? It would be useful to contextualize their method within the broader literature on posterior compression.

Minor Comments

Paper Structure and Readability

The current structure—presenting results and discussion before the methods—makes the paper challenging to follow, especially given its technical density and the mathematical formulation of the proposed sampling algorithms. Without a clear exposition of GSGHMC and HTE prior to their empirical evaluation and discussion, it is difficult to contextualize the core contributions or understand how specific hyperparameter studies relate to the underlying dynamics. While the reverse ordering may be justified for narrative reasons (e.g., to emphasize empirical gains), the technical nature of the work suggests that a conventional structure placing methods before results might better support clarity. Of course, this may be a matter of personal preference, but I feel that the paper would benefit from a more standard structure.

(Remarks on code availability)

I briefly inspected the repository linked in the manuscript and confirm that it appears to contain the relevant scripts and experimental configurations. However, I have not attempted to run the code myself and therefore cannot comment on its completeness or ease of use.

Version 1:

Reviewer comments:

Reviewer #1

(Remarks to the Author)

- Although Table 4 provides the number of epochs, it is difficult to compare the methods and understand when each should be used. For example, in the LeNet experiment, VOGN takes 500 epochs to achieve 16.2% MCE, while GSGHMC takes 5000 epochs to achieve 14.4% MCE. It would be clearer to present the performance under the same compute budget or the budget required to achieve the same performance.

- Regarding the bias, I suggest providing a more explicit characterization. For instance, the distance between the asymptotic distribution and the target distribution, as well as the dependency of the bound on the critical hyperparameters, would make the analysis more transparent.

- Similarly, a more explicit characterization of the convergence rate would be helpful. For example, the distance between the estimated distribution at step t and the target distribution. It is unclear how Section 4.4 and Equation 10 provide convergence guarantees, and further clarification would strengthen the argument.

- My concern regarding the positioning of this paper remains unresolved. It is unclear how the proposed methods differ from existing ones, both in theory and in experiments. The authors suggest that the paper's goal is to provide an empirical comparison in deep learning. If this is the case, then the paper overlooks a substantial body of literature in the area of uncertainty estimates in deep learning, which should be discussed and compared, such as [1,2,3,4].

Summary:

My concerns regarding error bars and comparisons to mini-batch-based Metropolis-Hastings have been addressed. However, other issues remain.

[1] Laplace Redux -- Effortless Bayesian Deep Learning, 2021

[2] A Simple Baseline for Bayesian Uncertainty in Deep Learning, 2019

[3] Cyclical Stochastic Gradient MCMC for Bayesian Deep Learning, 2020

[4] Dropout as a Bayesian Approximation: Representing Model Uncertainty in Deep Learning, 2016

(Remarks on code availability)

Reviewer #3

(Remarks to the Author)

The paper proposes two methods for approximate Bayesian inference in large machine learning models.

The proposed methodology seems sound, and the comparisons to contending methods are in my mind convincing.

My main objection with the paper is mainly the structure, though I understand that this is related to the journal guidelines. In particular the results before methodology is confusing as the reader lacks knowledge of what is really compared. Further, what I assume also are journal constraints limits the exposition of the proposed methods, and in particular gives the reader very few details about how the methods really function (which would be helpful for someone implementing the methods). I.e. how does changes to tuning parameters influence the behavior of the chains, where are the potential for further work further improving the methods and so on. Going directly to large scale comparisons hides such fine detail.

Minor details:

Line 130: missing reference

(Remarks on code availability)

No time for a proper code review

Version 2:

Reviewer comments:

Reviewer #3

(Remarks to the Author)

I'm happy with the revision. Don't think there are further things to do given the constraints of the journal.

(Remarks on code availability)

Point-by-point Response

Reviewer Nr. 1

Results

Strengths

- The proposed methods demonstrate improvements over existing approaches in terms of both accuracy and uncertainty estimation.
- The authors conduct an extensive ablation study to assess the effect of hyperparameters.

Weaknesses

- Many experimental results lack error bars, including Tables III, IV, and V. Since the performance of deep neural networks can be influenced by random seed selection, it is crucial to include error bars to assess whether the reported improvements are statistically significant.

Response R1.0: Thanks for raising this important concern. We fully agree with the need of seeded repetitions and accordingly extended table 4 with repetitions on 5 seeds, reporting the statistics by mean and standard deviation. We also include SGHMC in table 4 to demonstrate the difference between SGHMC and GSGHMC. Running multiple seeds during the hyperparameter study was not possible due to the enormous cost of the evaluation in general. However, to our knowledge there are no reported cases on overfitting to a specific seed and the comparison remains fair.

- The paper lacks a comparison to mini-batch-based Metropolis-Hastings methods such as AustereMH [1] and MHminibatch [1]. Especially, AustereMH has been used with stochastic gradient MCMC as shown in the original paper. Since AMAGOLD is asymptotically unbiased and the proposed methods appear to be asymptotically biased, these methods, which are asymptotically biased, still serve as reasonable baselines for comparison.

Response R1.1: Thanks for this valuable suggestion. While we could not include the method for all experiments, we extended its discussion from lines 765-782. In essence, the variable acceptance cost with an additional while loop over the data proposes a serious drawback for Bayesian deep learning. Therefore, it was not applied on Bayesian deep learning yet and remains a theoretically close, but empirically less established baseline.

- A cost comparison in the experiments would be valuable to better understand the trade-offs between computational efficiency and performance.

Response R1.2: Thanks for this suggestion. We include this by reporting the average number of total epochs per experiment in table 4. While the duration of an epoch is fixed on a hardware setup, this enables to compare independently of hardware requirements. This is additionally discussed in lines 450-452, 497-499 and 564-566.

Methods

Strengths

- The proposed methods are intuitive and easy to implement in practice.

Weaknesses

- It is unclear whether the proposed GSGHMC method converges to the target distribution asymptotically. Although the authors claim "bias-free sampling" in the abstract, my understanding is that the method introduces bias due to the use of mini-batching in the acceptance step. If this is the case, it would be beneficial to analyze the magnitude of the asymptotic bias. A similar concern applies to the proposed Hamiltonian Trajectory Ensemble (HTE).

Response R1.3: Thanks for this valuable suggestion. While it was not our intention to mix up terminology we stressed the sampling bias for our methods in the abstract to prevent wrong expectations in lines 31, 32 and 39. Additionally, we included the bias in the bullet point list (introduction, line 151). We expanded the discussion to equation (8) showing the bias of GSGHMC with its dependencies on the parameter c_n and the batch size. We extend the discussion regarding the bias of HTE in lines 921-924 and 931-944. In theory, we should not be able to quantify the bias of HTE without additional assumptions on the gradient noise (cp. [5]) since the divergence of the trajectories is required. A side effect of the regularization through gradient noise is a better generalizing model, which is in line with our observation of improved model performance and seems to be conveyed to the HTE method.

- The novelty of the proposed methods compared to existing approaches is not clearly established. Mini-batching or subsampling in the Metropolis-Hastings step is a well-known technique (e.g., [1, 2, 3, 4]). It remains unclear what distinguishes this work from prior methods. Providing a convergence rate analysis would strengthen the theoretical contributions of the paper, as such guarantees are commonly expected for MCMC-based methods.

Response R1.4: Thanks for raising these important concerns. We extend the discussion on the current literature from lines 765-782 and additionally stress the choice of our methods. We already covered [1, 2, 3] in brief before and now extended this. The convergence of GSGHMC is addressed in section 4.4 and equation (10). The necessary theorems are noted in the appendix. For HTE this issue is addressed in the changes we applied at lines 921-924 and 931-944. Furthermore, we note that the main contributions of this paper are in the experimental evaluation of methods in the field of deep learning rather than presenting a new baseline for any Bayesian inference problem. We argue, that these tasks require a special treatment barely recognized in the literature so far.

Summary

The experimental results suggest promising improvements over existing methods. However, key concerns remain regarding the statistical significance of these improvements, the novelty of the proposed approaches, and their theoretical properties. Addressing these issues would improve the strength and clarity of the paper.

Reviewer Nr. 2

Reviewer 2 (Remarks to the Author):

Summary

The paper presents two novel Metropolis-Hastings-based methods to improve sampling efficiency in Bayesian neural network (BNN) inference using stochastic gradient Hamiltonian Monte Carlo (SGHMC).

The motivation arises from the computational impracticality of exact Hamiltonian Monte Carlo (HMC) for large-scale deep learning models. The authors aim to address the trade-off between inference accuracy, uncertainty calibration, and computational efficiency by incorporating approximate Metropolis-Hastings corrections into SGHMC dynamics.

The first proposed method, Generalized SGHMC (GSGHMC), introduces minibatch-based acceptance steps in stochastic gradient-driven simulations, approximating the true posterior with tempered acceptance criteria while maintaining detailed balance through a reversible integrator (OBABO). This method is grounded in a theoretical formulation extending reversible Markov Chain Monte Carlo (MCMC) with stochastic gradients, relying on batch-tempered acceptance schemes with guarantees on convergence and local mode preservation under smoothness and regularity assumptions.

The second method, Hamiltonian Trajectory Ensemble (HTE), forgoes unbiased sampling in favor of long stochastic trajectories and Metropolis-like selection steps after deterministic integration with zero-temperature OBABO dynamics. While it does not converge to the true posterior, it produces ensembles with strong mixing properties and high predictive accuracy.

The authors benchmark both methods against SGHMC, AMAGOLD, and classical Bayesian neural network baselines (variational inference, variational online Gauss-Newton, and deep ensembles) on synthetic Gaussian benchmarks and real image classification tasks (LeNet/EMNIST and ResNet/CIFAR-10). They show that GSGHMC achieves the best calibration (ECE down to 0.010) while HTE achieves the highest accuracy (up to 93.7% on EMNIST), outperforming existing sampling-based methods. They also demonstrate that ensemble performance plateaus with significantly reduced ensemble sizes (down to 20% of original), thereby reducing inference costs without sacrificing accuracy.

The work demonstrates that with properly tuned Metropolis-Hastings acceptance mechanisms and careful integration of stochastic gradients, one can achieve scalable, high-performing BNN inference pipelines without the prohibitive computational burden of full HMC, while retaining or improving uncertainty calibration and predictive accuracy.

Comments

Calibration Trade-offs in HTE: Table V shows that the Hamiltonian Trajectory Ensemble (HTE) method achieves the highest accuracy across both LeNet/EMNIST and ResNet/CIFAR-10 settings, but this comes at the cost of degraded calibration. While the Expected Calibration Error (ECE) for HTE (0.027 for LeNet, 0.047 for ResNet) remains comparable to deterministic baselines, it is consistently worse than all other Bayesian approaches evaluated. Given that HTE does not converge to the true posterior, it is plausible that the increased accuracy derives from overconfident but miscalibrated predictions concentrated around a subset of low-energy modes explored through long stochastic trajectories.

Did the authors investigate whether this drop in calibration is systematic across all classes, or is it localized to particular regions of the input space? Additionally, were any calibration-improving post-processing techniques (e.g., temperature scaling) considered for HTE, and if so, what impact did they have on the observed trade-off between calibration and accuracy? How about out-of-distribution performance?

Response R2.0: Thanks for this careful observation. In response to R2.0, we expanded our evaluation by seeded repetitions and now report aggregated results. The aggregated results do not show the clear trade-off between accuracy and calibration anymore. We did run temperature scaling for all experiments, but found the results to change marginally keeping the non-scaled predictions. Additionally, we extend our results on classification tasks by an out-of-distribution evaluation in figure 4 and text changes were applied 501-523. In conclusion, the ResNet-HTE configuration still performed worse in terms of calibration error compared to the ResNet-SGHMC configurations, even when analyzing the results

with decreasing step-sizes. We discuss these observations in lines 586-593 and refer to the log-likelihood graph shown in the appendix visualizing the differences in the two models. Due to the extremely variable log-likelihood in the ResNet example, it is very likely to end up in different minima rather than carefully exploring those.

Task and Architecture Generality: The empirical results presented in the paper focus exclusively on image classification tasks using convolutional neural networks (CNNs), specifically LeNet on EMNIST and a small ResNet variant on CIFAR-10. While these benchmarks are standard, the restriction to vision-based classification is a limiting factor, especially given the paper’s strong emphasis on empirical validation. The architectural choices exclude evaluation on non-convolutional models, such as transformers, and no experiments are reported on regression or structured output tasks, where calibration and uncertainty quantification play an equally critical role.

Given that stochastic gradient-based MCMC methods are broadly applicable, it would be informative to test GSGHMC and HTE in alternative settings—such as time series forecasting, tabular regression, or even natural language tasks—to assess their robustness to domain shift, heteroscedastic noise, and varying data modalities. Were any such evaluations attempted or preliminary results obtained in non-image domains?

Response R2.1: Thanks for this valuable suggestion. We added a time-series prediction problem to the final evaluation, which is a part of table 4 and is introduced in lines 454-483 and discussed in lines 484-491 and 594-597. While we could not run all hyperparameter evaluation steps with this problem too, we show the general applicability in a different domain.

Acceptance Rate Dynamics and Interpretation: The paper reports an average acceptance rate of approximately $50\% \pm 3.5\%$ for both the GSGHMC and HTE methods during experiments (Section II-C). While this figure reflects a moderate trade-off between exploration and rejection, it looks surprisingly low for methods based on SG-MCMC. A detailed temporal analysis of how the acceptance rate evolves during burn-in and sampling would be valuable—e.g., does the rate stabilize, decline, or exhibit oscillatory behavior? Furthermore, the relationship between acceptance rate and predictive metrics (e.g., accuracy, NLL, calibration) is not explicitly addressed.

Is there evidence that lower acceptance rates correlate with more informative exploration of wide minima, or do they instead reflect miscalibrated proposals due to aggressive step sizes? Were different acceptance thresholds tested, or were diagnostics (e.g., autocorrelation, effective sample size) computed to assess mixing quality as a function of acceptance behavior over time?

Response R2.2: Thanks for pointing out these questions. We show the acceptance rate dynamics with uncertainties in figure C1 as well as autocorrelation for the test cases in figure C2. The text of Appendix C discusses their relevance in detail. We address this issue in the main document from lines 401-414. Our main observation regarding the acceptance rates was, that it did not change much, while the performance metrics did change a lot. Specifically for the step-size, practical experience suggest to decrease the step size for increasing the acceptance rate. However, this procedure was not applicable for large-scale deep learning models increasing the burn-in phase enormously or even preventing convergence to compatible minima. Therefore, we did not rely on acceptance rates for the tuning. We were optimizing performance while making sure to converge in reasonable time not getting stuck at local minima. During our hyper-parameter study, the acceptance rate was only minimally influenced.

On the Role of Subsampling and Ensemble Compression: The greedy subsampling procedure proposed in Section II-D, while practical, appears conceptually orthogonal to the core contributions of the paper. The main message of the work centers on improving the fidelity and efficiency of posterior sampling via novel Metropolis-Hastings corrections in stochastic gradient settings. Subsampling an ensemble post hoc for test-time speedup is a separate compression strategy that could in principle be applied to any Bayesian ensemble—regardless of how it was generated—and thus may not align tightly with the specific methodological innovations of GSGHMC or HTE. The observation that performance remains stable even when reducing ensembles by 60–80% is interesting, but does not inform the efficacy of the sampling method itself.

Moreover, there exist more principled approaches for compressing Bayesian predictive distributions, starting with Bayesian Dark Knowledge (Korattikara et al., 2015). Did the authors consider or compare such model distillation approaches to their subsampling baseline? It would be useful to contextualize their method within the broader literature on posterior compression.

Response R2.3: Thanks for pointing out this break in the storyline. Based on your concerns, we moved this section to 2.5 and rewrote lines 532-542, now emphasizing that these results are not completely orthogonal. We adapted the discussion in lines 619-633 accordingly. We changed the plot to show the effect on calibration as well, while the examples were moved to the appendix. We added your proposed approach to explicitly state that this evaluation is not meant to replace sampling chains or be used as a general method for ensemble reduction (line 538). We rather wanted to include it as an implicit limitation of our study that mainly focused the hyperparameter tuning on the accuracy, which is a common choice, but might miss some of the advantages of Bayesian sampling.

Minor Comments

Paper Structure and Readability: The current structure—presenting results and discussion before the methods—makes the paper challenging to follow, especially given its technical density and the mathematical formulation of the proposed sampling algorithms. Without a clear exposition of GSGHMC and HTE prior to their empirical evaluation and discussion, it is difficult to contextualize the core contributions or understand how specific hyperparameter studies relate to the underlying dynamics. While the reverse ordering may be justified for narrative reasons (e.g., to emphasize empirical gains), the technical nature of the work suggests that a conventional structure placing methods before results might better support clarity. Of course, this may be a matter of personal preference, but I feel that the paper would benefit from a more standard structure.

Response R2.4: Thanks for the feedback regarding the structure, however, the journal’s guidelines to authors are very explicit regarding the prescribed structure of manuscripts. To make the general structure more easy to overview, we included a short introductory paragraph (lines 170-176) to the Results section that the reader can choose whether to jump to methods first or stay with the results.

Remarks on code availability: I briefly inspected the repository linked in the manuscript and confirm that it appears to contain the relevant scripts and experimental configurations. However, I have not attempted to run the code myself and therefore cannot comment on its completeness or ease of use.

References

- [1] Anoop Korattikara, Yutian Chen, and Max Welling. Austerity in MCMC land: Cutting the Metropolis-Hastings budget. In International Conference on Machine Learning, 2014
- [2] Daniel Seita, Xinlei Pan, Haoyu Chen, and John Canny. An efficient minibatch acceptance test for Metropolis-Hastings. Uncertainty in Artificial Intelligence, 2017
- [3] Zhang, Ruqi, A. Feder Cooper, and Christopher De Sa. Asymptotically Optimal Exact Minibatch Metropolis-Hastings. Advances in Neural Information Processing Systems, 2020
- [4] Dougal Maclaurin and Ryan Prescott Adams. Firefly Monte Carlo: Exact MCMC with subsets of data. In Twenty-Fourth International Joint Conference on Artificial Intelligence, 2015
- [5] Michael Betancourt. The Fundamental Incompatibility of Scalable Hamiltonian Monte Carlo and Naive Data Subsampling. Proceedings of the 32nd International Conference on Machine Learning, 2015

Point-by-point Response

Reviewer Nr. 1

- Although Table 4 provides the number of epochs, it is difficult to compare the methods and understand when each should be used. For example, in the LeNet experiment, VOGN takes 500 epochs to achieve 16.2% MCE, while GSGHMC takes 5000 epochs to achieve 14.4% MCE. It would be clearer to present the performance under the same compute budget or the budget required to achieve the same performance.

Response: Thank you for this feedback. Sampling algorithms are naturally more costly than variational inference or even maximum a posteriori optimization. However, they come with the advantage of less model assumptions and the possibility to approximate multi-modal distributions. We conducted an additional study reported as appendix G of the manuscript, which among others (see below) analyzes convergence properties of the proposed methods. We refer to the study in lines 624-627 of the manuscript as follows: “Our convergence analysis on a small benchmark problem showed rapid convergence for medium and large step sizes and multi modal distributions (cp. Appendix G). In this analysis, the proposed GSGHMC and HTE methods show improved mixing over their competitors.” Although the study only compares sampling algorithms, the results provide more insights into the additional costs caused by sampling. We agree, that the cost is therefore hard to compare, but this remains a problem of Bayesian inference with sampling in general.

- Regarding the bias, I suggest providing a more explicit characterization. For instance, the distance between the asymptotic distribution and the target distribution, as well as the dependency of the bound on the critical hyperparameters, would make the analysis more transparent.
- Similarly, a more explicit characterization of the convergence rate would be helpful. For example, the distance between the estimated distribution at step t and the target distribution. It is unclear how Section 4.4 and Equation 10 provide convergence guarantees, and further clarification would strengthen the argument.

Response: Thanks for pointing this out. We added a further experiment to demonstrate this behavior in the new appendix G, covering the concerns from both points above. Our results allow for comparing the convergence in kernel stein discrepancy as a measure of distributional distance. Furthermore, it highlights the effect of different step sizes on this convergence.

- My concern regarding the positioning of this paper remains unresolved. It is unclear how the proposed methods differ from existing ones, both in theory and in experiments. The authors suggest that the paper’s goal is to provide an empirical comparison in deep learning. If this is the case, then the paper overlooks a substantial body of literature in the area of uncertainty estimates in deep learning, which should be discussed and compared, such as [1, 2, 3, 4].

Response: We agree that the suggested methods are a valuable addition to our study and now additionally report results on [2, 3, 4] in Table 4. The Laplace approximation of [1] is structurally similar to the VOGN results we already report and VOGN has been demonstrated to outperform Laplace in [5]. Therefore, we decided to not add [1] and keep the results of the VOGN method. We extended the description in lines 458 to 460 accordingly.

Summary: My concerns regarding error bars and comparisons to mini-batch-based Metropolis-Hastings have been addressed. However, other issues remain.

Reviewer Nr. 3

The paper proposes two methods for approximate Bayesian inference in large machine learning models. The proposed methodology seems sound, and the comparisons to contending methods are in my mind convincing.

My main objection with the paper is mainly the structure, though I understand that this is related to the journal guidelines. In particular the results before methodology is confusing as the reader lacks knowledge of what is really compared. Further, what I assume also are journal constraints limits the exposition of the proposed methods, and in particular gives the reader very few details about how the methods really function (which would be helpful for someone implementing the methods). I.e. how does changes to tuning parameters influence the behavior of the chains, where are the potential for further work further improving the methods and so on. Going directly to large scale comparisons hides such fine detail.

Response: Thanks for this feedback. We modified the introductory paragraph of the results section hinting towards the corresponding sections to account for this feedback (lines 170–174)

Minor details: Line 130: missing reference

Response: We fixed the reference in line 130.

References

- [1] Daxberger, Erik A., Agustinus Kristiadi, Alexander Immer, Runa Eschenhagen, M. Bauer and Philipp Hennig. “Laplace Redux - Effortless Bayesian Deep Learning.” *Neural Information Processing Systems* (2021)
- [2] Maddox, Wesley J., T. Garipov, Pavel Izmailov, Dmitry P. Vetrov and Andrew Gordon Wilson. “A Simple Baseline for Bayesian Uncertainty in Deep Learning.” *Neural Information Processing Systems* (2019).
- [3] Zhang, Ruqi, Chunyuan Li, Jianyi Zhang, Changyou Chen and Andrew Gordon Wilson. “Cyclical Stochastic Gradient MCMC for Bayesian Deep Learning.” *International Conference on Learning Representations*, (2020)
- [4] Gal, Yarin and Zoubin Ghahramani. “Dropout as a Bayesian Approximation: Representing Model Uncertainty in Deep Learning.” *International Conference on Machine Learning* (2015).
- [5] Osawa, Swaroop et al. “Practical deep learning with Bayesian principles.” *Conference on Neural Information Processing Systems* (2019)